# Emerging Multi-AI Agent Framework for Autonomous Agentic AI Solution Optimization

**Kamer Ali Yuksel & Hassan Sawaf**
aiXplain Inc., San Jose, CA, USA
{kamer, hassan}@aixplain.com

## Abstract

Agentic AI systems automate complex workflows but require extensive manual tuning. This paper presents a framework for autonomously optimizing Agentic AI solutions across industries, such as NLG-driven enterprise applications. It employs agents for Refinement, Execution, Evaluation, Modification, and Documentation, using iterative feedback loops powered by an LLM (*Llama 3.2-3B*). The system optimizes configurations without human input by autonomously generating and testing hypotheses, enhancing scalability and adaptability. Case studies demonstrate a significant boost in output quality, relevance, and actionability. Data, including original and evolved agent codes and outputs, are open-sourced.

## 1 Introduction

Agentic AI systems, composed of specialized agents collaborating on complex objectives, transform industries by automating decision-making and optimizing workflows. However, their optimization remains challenging due to intricate agent interactions and reliance on manual configurations. Advancements in large language models (LLMs) enable automated refinement of these systems. This paper presents an LLM-powered framework for optimizing Agentic AI systems through feedback loops. The approach autonomously refines agent configurations using qualitative and quantitative metrics, enhancing efficiency and scalability. Designed for enterprise deployment, it addresses workflow optimization issues in real-world.

Prior research explored various aspects of agentic system optimization and LLM integration. Huang et al. (2024) introduced *MLAgentBench*, evaluating language agents across tasks but focusing on performance rather than iterative refinement. Smith et al. (2023) examined *Large Model Agents (LMAs)* for inter-agent cooperation, aligning with this study's emphasis on autonomous refinement. Johnson & Liu (2023) demonstrated how LLMs enhance agent role optimization. Pan & Zhang (2024) proposed automated evaluators for refining agent performance but lacked scalability. Wang et al. (2024) introduced LLM-based skill discovery, resonating with this study's iterative task proposals. Hu et al. (2024) focused on modular components for planning, whereas this work emphasizes continuous refinement. Masterman et al. (2024) reviewed AI agent architectures, highlighting modularity and scalability, core aspects of this framework. Mitra et al. (2024) explored synthetic data generation for refining agent outputs. Yu et al. (2024) introduced the *Reflective Tree* for multistep decision-making, aligning with this study's iterative design. Pan et al. (2024) addressed feedback loop risks, which this framework mitigates via robust evaluation. Miller et al. (2024) emphasized agent benchmarks, reinforcing our focus on qualitative metrics. Lastly, Automated Design of Agentic Systems (ADAS) (Hu et al., 2024) focused on creating novel agents through a meta-agent.

In contrast, this study optimizes existing systems via iterative LLM-driven feedback loops, refining agent roles, tasks, and workflows for adaptability and scalability in dynamic environments. Our method leverages LLM-driven feedback loops for scalable, domain-independent optimization, surpassing prior methods reliant on predefined tasks. LLMs autonomously generate and evaluate hypotheses, iteratively improving agent roles and workflows with minimal human oversight. Case studies demonstrate the framework's scalability, adaptability, and effectiveness in overcoming domain-specific challenges. **Key contributions:** (1) Evolutionary adaptation of agent configurations (2) Autonomous refinement via feedback loops (3) Empirical results demonstrating improvements.

## 2 METHODOLOGY

Our method autonomously refines and optimizes Agentic AI systems through iterative cycles, continuously improving agent roles, goals, tasks, workflows, and dependencies based on qualitative and quantitative evaluations. The system is scalable across industries, with LLM-driven feedback loops ensuring adaptability to various NLG applications. Optimization follows two core frameworks: the Synthesis Framework and the Evaluation Framework. The Synthesis Framework generates hypotheses based on output analysis, where the Hypothesis and Modification Agents refine agent roles and tasks. The Evaluation Framework tests these refinements, ensuring continuous improvement.

The method begins with a baseline Agentic AI system, assigning agent roles, tasks, and workflows. An initial execution establishes a performance benchmark, analyzed using predefined evaluation metrics. The Hypothesis Agent suggests modifications based on this evaluation, which the Modification Agent implements by adjusting agent logic, workflows, and dependencies. The Execution Agent then runs these modifications, collecting performance metrics for analysis. The Evaluation Agent assesses clarity, relevance, and execution efficiency. The Selection Agent compares outputs, ranking variants to determine the most effective configuration. The Memory Module stores the best-performing iterations, enabling continuous refinement until predefined criteria are met.

The Refinement Agent orchestrates the optimization process, evaluating agent outputs against qualitative and quantitative criteria to identify areas for improvement. It utilizes metrics such as clarity, relevance, depth of analysis, and actionability to propose enhancements. The Hypothesis Generation Agent formulates modifications based on evaluation feedback, identifying inefficiencies in task delegation and role assignments. The Modification Agent applies these hypotheses by adjusting agent logic, modifying workflows, or altering dependencies. Each new system variant is stored with documented expected improvements. The Execution Agent runs these variants, ensuring agents perform as specified while tracking qualitative and quantitative outputs for evaluation. The Evaluation Framework assesses system outputs, leveraging an LLM to evaluate clarity, relevance, and execution metrics. The Evaluation Agent ensures alignment with system objectives and provides a detailed performance analysis. After each iteration, the Selection Agent ranks system variants based on evaluation scores, storing the top-performing configurations.

The refinement process begins with initializing a baseline configuration $C_0$ and generating an output $O_{C_0}$. Performance is evaluated using criteria such as clarity and relevance, producing an initial score $S(C_0)$. At each iteration $i$, hypotheses $\mathcal{H}_i$ are generated based on output evaluations and applied to produce a refined variant $C_{i+1}$. The configuration is updated if the new score $S_{i+1}$ surpasses the best-known score. Iterations continue until improvement falls below a predefined threshold $|S_{i+1} - S_{\text{best}}| < \epsilon$ or a maximum iteration count is reached. Upon execution, the Evaluation Agent assesses system outputs using qualitative and quantitative metrics, identifying areas for refinement. The Hypothesis Generation Agent proposes modifications, which the Modification Agent implements. The selection agent executes, evaluates, and ranks these refined variants. The Memory Module retains top-performing configurations, ensuring continuous optimization. This iterative process refines workflows and enhances system performance, yielding an optimized Agentic AI.

## 3 CASE STUDIES

The refinement of agent systems is crucial to meeting evolving industry needs. This section presents case studies showcasing improvements in market research, AI architecting, career transitions, outreach, LinkedIn content, meetings, lead generation, and presentations. Each case highlights initial challenges, strategic modifications, and resulting improvements. The findings emphasize the role of specialization and data-driven decisions in enhancing performance. Case study data, including code, outputs, and evaluations, can be accessed at: https://anonymous.4open.science/r/evolver-1D11

The case studies below highlight the transformative impact of targeted modifications and specialized roles in agent systems. Each iteration improved alignment, accuracy, relevance, clarity, and actionability, addressing initial challenges and enhancing domain-specific utility. The experiments confirm the necessity of continuous refinement to meet evolving industry and user needs. Specialization and user-centric design significantly enhance output quality and effectiveness. Insights from Sulc et al. (2024) emphasize self-improving agents that autonomously adjust roles and interactions via feedback loops. Experiment results demonstrate the potential for dynamic adaptation, making

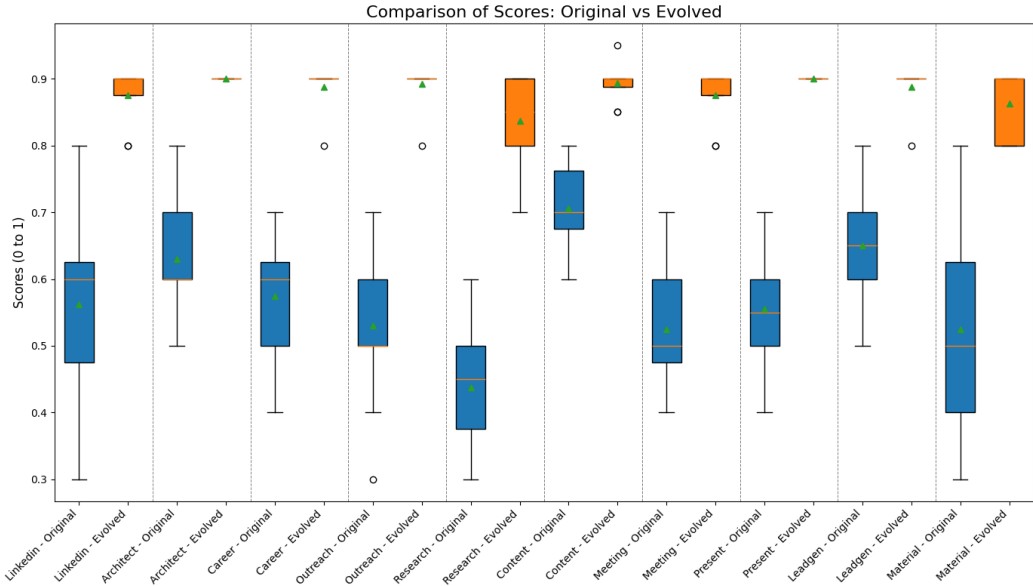

Figure 1: Original vs Evolved System Comparisons across Multiple Case Studies: Each pair of bars represents the evaluation scores for original and evolved systems, highlighting significant alignment, clarity, relevance, and actionability improvements achieved by refining agents, tasks, and workflows.

it ideal for environments with shifting objectives. The evolved systems consistently demonstrate higher scores, indicating the effectiveness of introducing specialized roles and targeted modifications. These findings highlight the transformative impact of iterative refinement, specialization, and adaptability in agent system design.

## EVALUATION KEY INSIGHTS

**Consistent Gains:** Evolved systems scored near or above 0.9, proving effectiveness.

**Reduced Variability:** More consistent, reliable outputs due to specialized roles.

**Targeted Enhancements:** Outreach, Market Research, and Medical AI agents showed the greatest improvements.

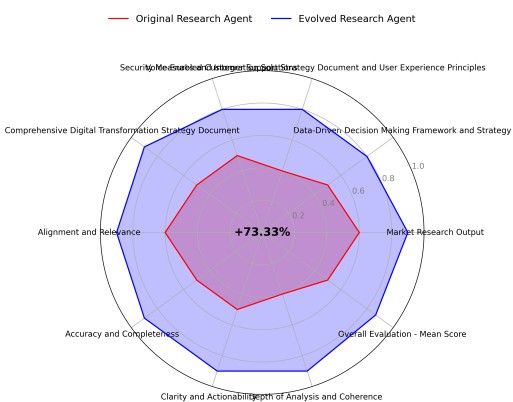

Figure 2: Market Research Agent Refinement

## 3.1 MARKET RESEARCH AGENT

The original agent lacked depth, strategy development, and output quality. The refined version introduced specialized roles—Market Analyst, Data Analyst, and UX Specialist—enhancing analysis, decision-making, and user-centered design. These improvements increased output relevance, accuracy, and clarity, achieving a 0.9 evaluation score.

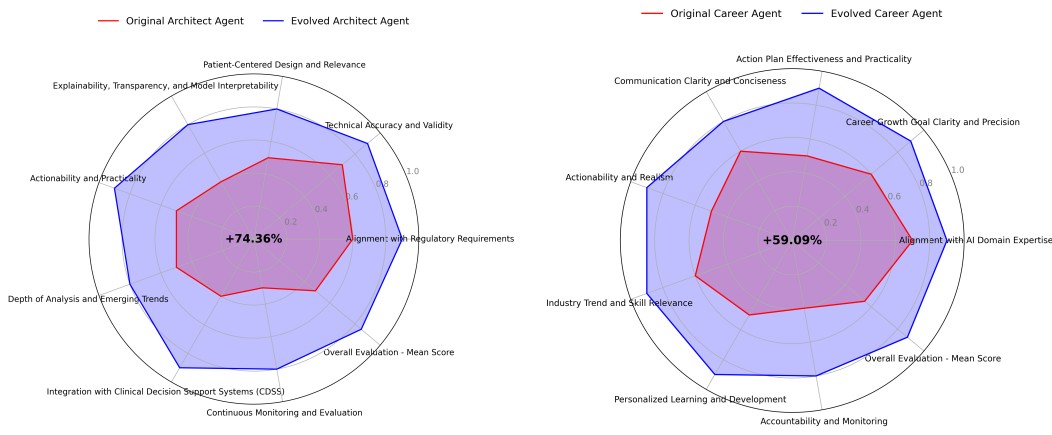

Figure 3: AI Architect Agent Refinement

Figure 4: Career Transition Agent Refinement

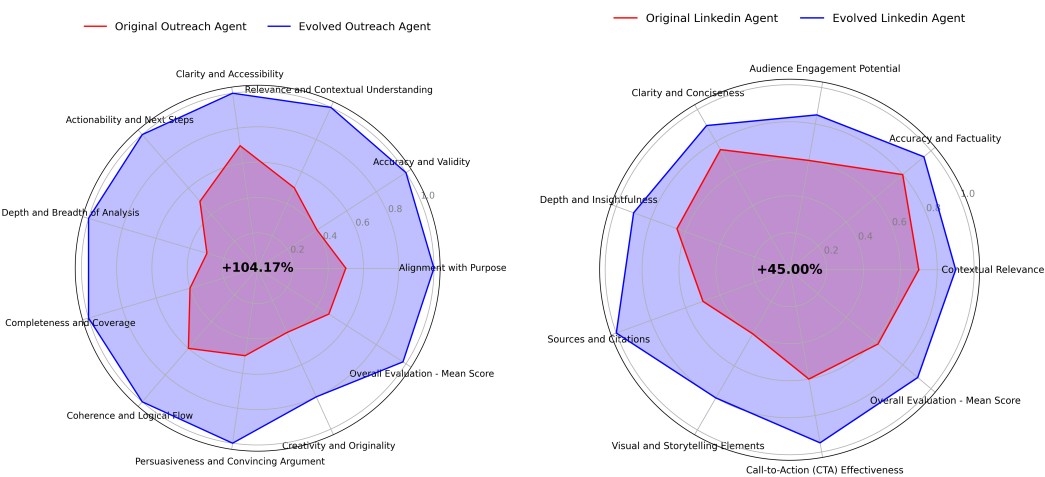

Figure 5: Outreach Agent Refinement

Figure 6: Career Transition Agent Refinement

## 3.2 MEDICAL AI ARCHITECT AGENT

Challenges in compliance, patient engagement, and explainability hindered effectiveness. The refined system incorporated a Regulatory Compliance Specialist and Patient Advocate, improving adherence to standards and transparency. The evolved system scored 0.9 in compliance, 0.8 in patient focus, and 0.8 in explainability.

## 3.3 CAREER TRANSITION AGENT

The career transition agent lacked alignment with industry expertise and clarity in career goals. Adding Domain Specialist and Skill Developer improved specificity, structured action plans, and clear timelines, achieving 0.9+ alignment and clarity.

## 3.4 OUTREACH AGENT

Initially, the outreach system was limited in scope and output quality. Adding specialized roles for supply chain analysis, optimization, and sustainability improved clarity, accuracy, and actionability, making it a valuable tool for e-commerce logistics.

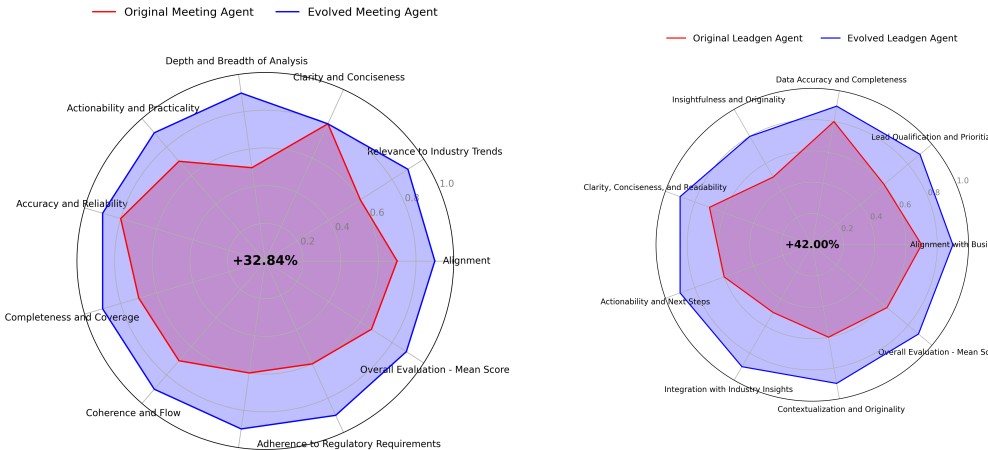

Figure 7: Meeting Agent Refinement

Figure 8: Lead Generation Agent Refinement

### 3.5 LINKEDIN AGENT

Early versions struggled with engagement and credibility. The refined agent, featuring an Audience Engagement Specialist, adopted a dynamic strategy with audience analytics, significantly enhancing relevance, clarity, and interaction potential.

### 3.6 MEETING AGENT

Originally misaligned with industry trends and lacking depth, the agent now integrates AI industry experts and regulatory specialists, significantly improving relevance and actionability, achieving 0.9+ scores across evaluation categories.

### 3.7 LEAD GENERATION AGENT

The initial system struggled with business alignment and data accuracy. Adding Market Analyst and Business Development Specialist roles improved lead qualification and data integrity, increasing alignment (91%) and accuracy (90%).

## 4 LIMITATIONS

LLM-based feedback and evaluation may introduce biases and inaccuracies. Its effectiveness depends on the generated or given criteria; poor criteria lead to suboptimal refinements. Lastly, the iterative optimization process is computationally intensive concerning the LLM inferencing load.

## 5 CONCLUSION

This paper presents a method for autonomous refinement and optimization of AI Agents. Leveraging iterative feedback loops, hypothesis generation, and automated modifications, the method enhances efficiency and effectiveness while minimizing human intervention. Its scalability and adaptability make it ideal for optimizing complex AI agents in large-scale applications. While demonstrating significant advancements, further exploration can refine its capabilities. Integrating human-in-the-loop strategies could balance autonomy and nuanced judgment, particularly in uncertain environments. Hybrid systems leveraging human expertise could enhance safety and reliability.

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
