# OpenReview forum: "Emerging Multi-AI Agent Framework for Autonomous Agentic AI Solution Optimization"
_ICLR.cc/2025/Workshop/AgenticAI — ICLR 2025 Workshop AgenticAI Poster_

### Official Review · Reviewer_TiMc · 2025-03-02
**EMERGING MULTI-AI AGENT FRAMEWORK FOR AUTONOMOUS AGENTIC AI SOLUTION OPTIMIZATION**

**Rating:** 6
**Confidence:** 5

**Review:**

1. Autonomous AI agent optimization framework using iterative LLM-driven refinement.

2. Eliminates manual tuning by automatically generating, evaluating, and modifying agent roles.

3. Uses specialized agents (Refinement, Execution, Evaluation, Modification, Documentation) for optimization.

4. Case studies in market research, AI architecting, career transitions, outreach, and lead generation show effectiveness.

5. Evolved systems outperform original versions, achieving higher clarity, relevance, and actionability scores.

6. Scalable across multiple industries but primarily tested on NLG applications.

7. Mitigates LLM feedback loop risks but still susceptible to LLM-induced biases and hallucinations.

8. No comparisons against strong baselines like AutoGPT, BabyAGI, or RL-based agent optimizers.

9. Computational cost of iterative LLM inferencing not addressed, raising efficiency concerns.

10. Framework’s adaptability to non-text-based AI agents (vision, robotics) remains untested.

11. Risk of reward hacking if LLM-generated evaluation criteria introduce self-reinforcing biases.

12. Lacks hybrid human-AI evaluation—unclear if human oversight could enhance refinement quality.

13. No statistical significance tests on performance gains, reducing confidence in generalizability.

14. Could be strengthened by adversarial scoring models to prevent deceptive optimizations.

15. Solid framework but needs efficiency validation and stronger empirical comparisons for ICLR acceptance.

---

### Official Review · Reviewer_WfTn · 2025-03-02

**Rating:** 3
**Confidence:** 4

**Review:**

The paper presents an LLM-powered framework for optimizing Agentic AI systems through feedback loops. The paper has performed various case studies for their proposed framework.

Weakness:
- While the paper discusses related work in L33-48, it should make it more clear how the proposed approach differs from previous work that also incorporates iterative design.
- The problem statement is not well introduced and could be framed more clearly to establish the motivation and significance of the study.
- The methodology lacks crucial details, such as the prompts used for different agents. The design principle behind the prompt is missing, and it remains unclear whether prompt engineering is required for every task.
- Regarding evaluation, it appears that the scores in Figure 1 are generated by the evaluation agent? If so, given that the backbone model is Llama 3.2-3B, it is difficult to assess the reliability of these scores. Incorporating additional standard metrics or human evaluation would strengthen the credibility of the results.
- The paper would also benefit from the evaluation of established benchmarks, such as MLAgentBench, by comparing the performance of the original and refined models.
- Text overlaps in Figure 2
- The paper claims that the proposed method enhances efficiency, but no experiments have been conducted to support this statement.

---

### Official Review · Reviewer_BDs1 · 2025-03-03
**Emerging Multi-AI Agent Framework for Autonomous Agentic AI Solution Optimization**

**Rating:** 7
**Confidence:** 4

**Review:**

# 1. Summary

This paper presents a framework that leverages large language models to autonomously refine and optimize agent-based AI systems for enterprise applications. Specialized agents—Hypothesis, Modification, Execution, Evaluation, and Documentation—collaborate in iterative feedback loops until the performance improvement

$$
|S_{i+1} - S_{best}| < \epsilon,
$$

falls below a threshold. The method enhances scalability, adaptability, and overall performance.

# 2. Strengths

1. **Innovative Integration:**
 The paper introduces a fresh approach by using LLM-driven feedback loops to let the system improve itself, resembling a team of experts that refines its own strategies over time.

 2. **Clear Methodological Design:**
 The framework clearly defines agent roles and iterative interactions with less technical complexity.

 3. **Empirical Support:**
 Case studies are based on real numbers and demonstrate specific improvements in major performance indicators.

# 3. Weaknesses
1. **High Computational Overhead**
 The iterative LLM feedback loop is resource-intensive, which may limit scalability

2. **Risk of Evaluation Biases:**
 Relying on LLM assessments could introduce biases that affect the reliability of the results.

---

### Decision · Program_Chairs · 2025-03-05

Accept (Poster)